# Development of Dibenzothiazepine Derivatives as Multifunctional Compounds for Neuropathic Pain

**DOI:** 10.3390/ph15040407

**Published:** 2022-03-27

**Authors:** Young-Hwan Jung, Yeo Ok Kim, Koon Mook Kang, Hyung Gon Lee, Borum Son, Xuehao Han, Eunseok Oh, Siwon Kim, Seon Hee Seo, Jong-Hyun Park, Ki Duk Park, Woong Mo Kim, Myung Ha Yoon, Yong-Chul Kim

**Affiliations:** 1School of Life Sciences, Gwangju Institute of Science and Technology (GIST), Gwangju 61005, Korea; young-hwan.jung@nih.gov (Y.-H.J.); km1944@gm.gist.ac.kr (K.M.K.); br219@hanmail.net (B.S.); 2Department of Anesthesiology and Pain Medicine, Chonnam National University Medical School, Gwangju 61469, Korea; yeok1017@hanmail.net (Y.O.K.); leehg@chonnam.ac.kr (H.G.L.); 752253335h@gmail.com (X.H.); kimwm@chonnam.ac.kr (W.M.K.); 3Center for Creative Biomedical Scientists, Chonnam National University Medical School, Gwangju 61469, Korea; 4Department of Chemistry, Gwangju Institute of Science and Technology (GIST), Gwangju 61005, Korea; dmstjr0130@snu.ac.kr; 5Convergence Research Center for Brain Science, Korea Institute of Science and Technology (KIST), Seoul 02792, Korea; h15512@kist.re.kr (S.K.); shseo@kist.re.kr (S.H.S.); jhyunprk@kist.re.kr (J.-H.P.); kdpark@kist.re.kr (K.D.P.); 6Division of Bio-Medical Science & Technology, KIST School, Korea University of Science and Technology, Seoul 02792, Korea; 7Department of Biomedical Science and Engineering, Gwangju Institute of Science and Technology (GIST), Gwangju 61005, Korea

**Keywords:** neuropathic pain, multifunctional compound, multiple mechanisms of action, tianeptine, opioid receptor, neurotransmitter transporter

## Abstract

Neuropathic pain is a chronic and sometimes intractable condition caused by lesions or diseases of the somatosensory nervous system. Many drugs are available but unfortunately do not provide satisfactory effects in patients, producing limited analgesia and undesirable side effects. Thus, there is an urgent need to develop new pharmaceutical agents to treat neuropathic pain. To date, highly specific agents that modulate a single target, such as receptors or ion channels, never progress to the clinic, which may reflect the diverse etiologies of neuropathic pain seen in the human patient population. Therefore, the development of multifunctional compounds exhibiting two or more pharmacological activities is an attractive strategy for addressing unmet medical needs for the treatment of neuropathic pain. To develop novel multifunctional compounds, key pharmacophores of currently used clinical pain drugs, including pregabalin, fluoxetine and serotonin analogs, were hybridized to the side chain of tianeptine, which has been used as an antidepressant. The biological activities of the hybrid analogs were evaluated at the human transporters of neurotransmitters, including serotonin (hSERT), norepinephrine (hNET) and dopamine (hDAT), as well as mu (μ) and kappa (κ) opioid receptors. The most advanced hybrid of these multifunctional compounds, **17**, exhibited multiple transporter inhibitory activities for the uptake of neurotransmitters with IC_50_ values of 70 nM, 154 nM and 2.01 μM at hSERT, hNET and hDAT, respectively. Additionally, compound **17** showed partial agonism (EC_50_ = 384 nM) at the μ-opioid receptor with no influence at the κ-opioid receptor. In in vivo pain animal experiments, the multifunctional compound **17** showed significantly reduced allodynia in a spinal nerve ligation (SNL) model by intrathecal administration, indicating that multitargeted strategies in single therapy could considerably benefit patients with multifactorial diseases, such as pain.

## 1. Introduction

Neuropathic pain progresses as the result of lesions or diseases affecting the somatosensory nervous system either in the periphery or central regions [1]. Neuropathic pain is now considered to be a distinct clinical entity, despite a large variety of etiologies, and is classified as a variety of disease entities, including painful polyneuropathy, postherpetic neuralgia, trigeminal neuralgia, and poststroke pain [2]. Neuropathic pain manifests as a chronic and sometimes intractable condition, which can adversely affect a patient’s overall health-related quality of life and consequently places a high economic burden on individuals and society [3,4]. Neuropathic pain still presents a major therapeutic challenge despite the considerable progress that has been made in understanding its mechanisms through a number of studies assessing the efficacy and safety of drugs used in symptomatic treatment [5].

Existing therapeutic approaches include the use of opioid, antidepressant, and anticonvulsant drugs [6]. The response rate for medication of neuropathic pain is approximately 30% to 50% analgesic effects in up to 50% of pain patients [7]. These approaches have limitations, including adverse events (e.g., nausea, constipation, dependence, and drug-drug interaction) and decreasing analgesic effects of tolerable doses over time [8].

To address unmet medical needs, paradigms for the treatment of multifactorial diseases, such as pain, have been shifted to the development of multifunctional drugs (those exhibiting two or more pharmacological activities in a single dosage form). The advantages of multifunctional compounds (MFCs) are (1) additive or synergistic therapeutic responses, (2) improved druggable characteristics, (3) improved predictability of PK and PD relationships, and (4) prolonged duration of effectiveness compared with drug combinations. Multifunctional compounds have been developed by pharmaceutical industries and academic institutes over the past decade. Some examples of successful MFC candidates for the treatment of CNS disorders are ziprasidone (**1**), duloxetine (**2**), tramadol (**3**), ladostigil (**4**) and M-30 (**5**) (Figure 1) [9,10]. Among these MFCs, duloxetine (a serotonin and norepinephrine reuptake inhibitor) and tramadol (a μ-opioid agonist, serotonin reuptake inhibitor, M1 and M3 muscarinic acetylcholine receptor antagonist, and TRPA1 inhibitor) are currently used as first- or second-line treatments for pain [11,12].

Tianeptine was discovered in the 1980s in research studies on developing new antidepressant drugs that would be more effective and safer than tricyclic antidepressants (TCAs). The molecular structure of tianeptine includes a substituted dibenzothiazepine nucleus and a long aminoheptanoic acid side chain. This seven-carbon amino acid [NH-(CH_2_)_6_-COOH] side chain distinguishes tianeptine from conventional TCAs that have distinct neurochemical properties. In contrast to classic TCAs and selective serotonin reuptake inhibitors (SSIRs), tianeptine has been shown to selectively enhance the synaptic uptake of serotonin [13], but not norepinephrine or dopamine, in rat-brain synaptosomes [14]. The involvement of glutamate in the mechanism of action of tianeptine is consistent with the well-developed preclinical literature [15]. The efficacy and tolerability of tianeptine in patients with depression have been studied extensively, and tianeptine has been clearly demonstrated to alleviate anxious symptoms associated with depression [15,16]. Recent experimental studies have indicated that tianeptine shows analgesic activity in pain animal models. In mice, intraperitoneal administration of tianeptine produced antinociceptive effects in tail-flick and hot plate tests [17], as well as in spinal-nerve-ligated and chemotherapy-induced neuropathic pain models [18]. Intrathecal administration of tianeptine has been found to induce analgesia in a rat model of neuropathic pain [19] and in an inflammatory pain model [20]. The mechanism for the analgesic effects of tianeptine has been associated with the agonisms of opioid receptors reported in the literature [21]. In the absence of disease-modifying or curative agents for the management of neuropathic pain, improved analgesic efficacy remains a primary unmet need. As some current analgesic medicines have been developed by repositioning antidepressant agents, [22] full structure-activity relationships (SARs) and optimization of tianeptine derivatives are potential strategies for developing novel analgesics with bifunctional activities at transporters and opioid receptors.

In this study, novel MFCs were investigated through the hybridization of key pharmacophores of current pain drugs with tianeptine with the aim of developing multitargeted antiallodynic agents (Figure 2). Novel hybrid compounds were successfully developed with potent in vitro functional activities at transporters of neurotransmitters and μ-opioid receptors, as well as with antiallodynic effects in spinal-nerve-ligated rats. Herein, we report the discovery of novel multitargeted antiallodynic agents with a clear mechanism of action developed from the structural conjugation of appropriate pharmacophores of various pain-related compounds with tianeptine.

## 2. Results and Discussion

### 2.1. Chemistry

The syntheses of the serotonin-conjugated tianeptine derivatives, compounds **7**, **11a**–**e**, **12** and **17** are shown in Figure 1, Figure 2, Figure 3 and Figure 4. To determine the optimum carbon chain length of the linker between the pharmacophore 3-chloro-6-methyl-6,11-dihydrodibenzo[c,f][1,2]thiazepine 5,5-dioxide of tianeptine and serotonin, we replaced the heptanoic acid group in tianeptine with alkanoic acids of various carbon chain lengths. Briefly, the chloride of the starting material, 3,11-dichloro-6-methyl-6,11-dihydrodibenzo[c,f][1,2]thiazepine 5,5-dioxide, was substituted with the free amine of serotonin or aminoalkyl esters **8a**–**e** to obtain **7** and **9a**–**e**. The carboxylic acids of tianeptine derivatives (**10a**–**e**), synthesized by the hydrolysis of **9a**–**e** using 5% aqueous NaOH, and tianeptine itself were coupled with the free amine of serotonin to yield **11a**–**e** and **12** (Figure 1, Figure 2 and Figure 3).

Compounds **13** and **14** were synthesized by conducting coupling reactions of tianeptine itself with 5-methoxytryptamine or tryptamine, respectively (Figure 3).

Figure 4 was an attempt at introducing an amine linker instead of an amide linker. A mild alkylation reaction of 7-aminoheptanol with compound **6** afforded compound **15**, which was subsequently subjected to a swan oxidation reaction for the synthesis of the aldehyde compound **16**. Finally, compound **17** was synthesized by reacting serotonin with compound **16** under the conditions for reductive amination using sodium cyanoborohydride.

Straight alkyl chain-linkered compounds **21** and **22** were synthesized as shown in Appendix A. 3-(2-propenyl)indole was used as the starting material for the cross metathesis reaction to afford compound **18**, of which the nitrile group was reduced to a primary amine moiety using LiAlH_4_ to yield compound **19**. Compound **20** was obtained from the hydrogenation reaction of the unsaturated double bond of compound **19** with Pd/C under H_2_ gas. Subsequently, substitution reactions were conducted for compounds **19** and **20** to yield compounds **21** and **22**, respectively.

The pregabalin structure was introduced by conjugating tianeptine with compound **23** to obtain compound **24**, which was subsequently hydrolyzed using 5% aqueous NaOH to yield compound **25** (Figure 3).

Last, fluoxetine and cyanoindole derivatives, compounds **26** and **27**, respectively, were synthesized by reductive amination with compound **16** (Figure 4).

### 2.2. Biological Activity

#### 2.2.1. In Vitro Assay for the Dibenzothiazepine Derivatives

The newly synthesized dibenzothiazepine derivatives were evaluated for their multifunctional activities, including the inhibition of neurotransmitter transporters, along with κ- and μ-opioid agonisms. GBR12909, fluoxetine and nisoxetine were examined together as positive controls for dopamine (DAT), serotonin (SERT) and norepinephrine (NET) transporters, respectively. Tianeptine itself displayed very weak inhibitory activities at all three transporters, as described in the introduction section, because of its mechanism of action for enhancing synaptic uptake of serotonin. The agonistic activities of compounds at κ- and μ-opioid receptors were evaluated against the corresponding antagonists dynorphin A (EC_50_ = 20 nM at the κ-opioid receptor) and DAMGO (EC_50_ = 53 nM at the μ-opioid receptor).

Table 1 shows that the inhibitory activities of the neurotransmitter transporters of the tianeptine derivatives (**7**, **11a**–**e** and **12**) were investigated to determine the optimized carbon chain length of the linker and the most important pharmacophores for the targets. Initially, compound **7**, a serotonin-substituted dibenzothiazepine derivative, showed weak but slightly improved inhibitory activities at the three transporters compared with tianeptine. Subsequently, the serotonin-conjugated tianeptine derivatives **11a**–**e** and **12**, for which the carbon chain lengths of the aliphatic acids varied from *n* = 1 to 6, were evaluated to optimize the distance between the two pharmacophores. Increasing the carbon chain length of the aliphatic acids resulted in an increased tendency toward hSERT inhibitory activities, whereas the inhibitory activities at hDAT and hNET were not changed. Among the series of derivatives, the 6-carbon-chain compound **12**, the derivative of serotonin conjugated with tianeptine itself, displayed dramatically potent hSERT inhibitory activity with an IC_50_ value of 258 nM and no appreciable activities at the other two transporters (HEK-hDAT, HEK-hNET and HEK-hSERT were provided by Professor Bryan Roth, University of North Carolina at Chapel Hill). Thus, the carbon chain length of the linker of **12** (*n* = 6) appeared to be the most adaptable length, considering the weak inhibition effects of compounds **7** and **11a**–**e** (*n* = 1–5) on the human serotonin transporter.

The importance of the 5–OH group of the indole moiety of compound **12** was investigated by synthesizing and testing compounds **13** and **14**, which have –OCH_3_ and –H groups at the 5-position of the indole, respectively. The results show that the hSERT inhibitory activities of compounds **13** and **14** were significantly decreased compared with those of compound **12**, suggesting that the corresponding binding site in hSERT may mainly be accessed by the interaction of the 5-hydroxyl group of the indole moiety as a hydrogen bonding donor (Table 1). However, the inhibitory activities of compound **14** at hDAT and hNET (IC_50_ = 3.81 μM and 65.2% inhibition at 10 μM, respectively) were increased compared with those of compound **12** (IC_50_ = 9.66 μM and 30.1% inhibition at 10 μM, respectively), indicating that the binding mode of the indole moiety at the dopamine and norepinephrine transporters may be different from that at the serotonin transporter.

Further SAR analysis of the amide linker resulted in compound **17** (IC_50_ = 70 nM) with a sec.-amine linker instead of an amide linker that showed 3-fold higher hSERT inhibitory activity than compound **12**. The inhibitory activities of compounds **21** and **22**, which have no nitrogen-containing decane and 2-decene linkers connected with indole groups, were even more reduced compared to those of compound **14**. These results indicate that the properties of the NH group in the linker, including hydrogen donor ability or basicity, critically affect inhibitory activity at hSERT, such that the corresponding sec.-amine linker in compound **17** could interact with the corresponding hSERT binding pocket more effectively than the amide or carbon linkers. Interestingly, compound **17** showed dramatically improved inhibitory activities at hDAT and hNET as well, with IC_50_ values of 2.01 μM and 0.154 μM, respectively, corresponding to the most potent inhibitory activities at hDAT or hNET among the compounds studied in Table 1. Therefore, the sec.-amine group linked between the pharmacophore of the tianeptine structure, and the serotonin moiety (compound **17**) is expected to play an important inhibitory role by interacting with all three transporters.

Substituents other than serotonin, such as pregabalin (**25**), which is a calcium channel blocker and analgesic, and fluoxetine (**26**) or 3-(2-aminoethyl)-1H-indole-5-carbonitrile (**27**), which are known serotonin reuptake inhibitors, were also linked with tianeptine derivatives and evaluated. Among these substituents, compound **27** showed good inhibitory activity at hSERT with an IC_50_ value of 462 nM, but compounds **25** (11.3% at 10 μM) and **26** (IC_50_ = 2.18 μM) displayed weak or lower inhibitory activities, suggesting that the corresponding binding pocket in hSERT may mainly be accessed based on the hydrogen bonding donor and acceptor abilities of the hydroxyl or cyano groups of the indole moiety of the tianeptine derivatives.

#### 2.2.2. In Vitro Assay for Opioid Receptors Agonisms

Gassaway et al. reported tianeptine as an efficacious μ-opioid receptor (MOR) agonist, with EC_50_ values of 194 nM at hMOR and 641 nM at mMOR in radioligand binding and cell-based functional assays, respectively [21]. Thus, the selected compounds were further evaluated for agonistic activities at κ- or μ-opioid receptors (Table 2). In the studied assay systems, tianeptine itself did not show appreciable biological activities at κ- or μ-opioid receptors. However, compounds **12** and **17**, the most potent neurotransmitter uptake inhibitors investigated in this study, showed partial agonisms with EC_50_ values of 1200 and 384 nM (corresponding to a 5.9-fold increase in Ca^2+^ efflux in HEK-hMOR), respectively, at the μ-opioid receptor compared to the well-known ligand DAMGO (EC_50_ = 53 nM), whereas compounds **12** and **17** did not show agonistic activities at the κ-opioid receptor. The other selected compounds, **14**, **21**, **22**, and **25**–**27**, were inactive or had lower potency than compounds **12** and **17** for agonistic activities at κ- or μ-opioid receptors. Based on these results, compounds **12** and **17** were further evaluated as multifunctional compounds in in vivo experiments.

#### 2.2.3. β-Arrestin Recruitment at the hOPRM1 Receptor

Manglik et al., reported that opiate-induced side effects, such as tolerance, respiratory depression, and constipation, were implicated in the recruitment of β-arrestin, which is a downstream signaling pathway for μ-opioid receptor activation [23]. Based on this report, selected compounds were investigated for their recruitment activities of β-arrestin (Table 3).

The most potent compound, **17** (EC_50_ = 1556 nM), showed a lower level of β-arrestin recruitment than compounds **12** (EC_50_ = 134 nM) and **14** (EC_50_ = 113 nM). All other compounds with low agonisms at the μ-opioid receptor, including tianeptine, showed no or weak recruitment activities for β-arrestin. Therefore, compound **17** can be used to address concerns about unmet medical needs regarding side effects for the treatment of neuropathic pain.

#### 2.2.4. In Vivo Behavioral Responses

Motor functions were assessed by examining righting and placing/stepping reflexes [24]. The former was evaluated by placing a rat on its back in a horizontal position on a table, which immediately resulted in coordinated twisting of the body into an upright position. The upright position was evoked by the rat drawing the dorsum of either hind paw across the edge of a table. Normally, rats attempt to place a paw into a position that enables walking. The pinna and corneal reflexes were evoked by stimulating the ear canal or cornea, respectively, with a string. Normally, rats spontaneously shake their heads or blink. Normality of behavior was classified as present or absent. To evaluate the behavioral changes induced by compounds **12** and **17**, additional rats were examined for 3 h after intrathecal administration of the highest doses of the agents used. The motor function and pinna and corneal reflexes were normal after intrathecal administration of compounds **12** and **17** at the highest doses. Overt abnormal behavioral changes were not observed.

#### 2.2.5. In Vivo Pharmacology of Tianeptine Derivatives in Neuropathic Pain Animal Models

The in vitro metabolic stabilities of compounds **12** and **17** were determined to be extremely low (Appendix A, with percent remaining after 30 min incubation with the S9 microsomal fraction). For this reason, we performed in vivo experiments at the spinal level by intrathecal administration of those two compounds. Intrathecal compound **12** increased the paw withdrawal threshold (PWT) compared to the vehicle in spinal-nerve-ligated neuropathic pain rats (Figure 3). In addition, intrathecal compound **17** increased the PWT (Figure 4). The maximum possible effect (MPE) of compounds **12** and **17** were 73% and 83% at the highest dose, respectively.

#### 2.2.6. Multiple Mechanisms of Action for Treatment of Neuropathic Pain

We successfully developed multifunctional compounds to treat neuropathic pain that have multiple mechanisms of action, such as neurotransmitter transporter inhibition and μ-opioid agonism. The significance of the study results is as follows. (1) The mechanism of action of tianeptine as a serotonin reuptake enhancer is reversed in the treatment of neuropathic pain. The uncertainty in the mechanism was resolved by developing multifunctional compounds that act as a serotonin reuptake inhibitor. (2) The major mechanisms of action of tramadol are neurotransmitter-transporter inhibition and μ-opioid agonism. The efficacy of tramadol for the treatment of neuropathic pain has been proven by clinical experiences of neuropathic pain patients. Interestingly, the representative multifunctional compound such as tramadol targeting multiple pathways of neuropathic pain, **17**, showed antiallodynic effects at the spinal level. These results suggest that multifactorial diseases, such as pain, could be modulated efficiently because of the aforementioned advantages of the investigated multifunctional compounds.

## 3. Materials and Methods

### 3.1. Chemical Synthesis

^1^H and ^13^C NMR spectra were obtained on JEOL JNM-ECX 400P spectrometer (Tokyo, Japan) at 400 MHz (^1^H-NMR) and 100 MHz (^13^C-NMR), and spectra were taken in CDCl_3_, DMSO-*d*_6_ or CD_3_OD. Data are reported as follows: chemical shift, integration, multiplicity (s, singlet; d, doublet; t, triplet; m, multiplet; br, broad). Coupling constants (*J*) are reported in Hz. Purification by column chromatography was carried out using precoated thin-layer silica gel plates (MERCK silica gel 60; F254, 0.040−0.063 mm). Melting point was determined with a Büchi apparatus (M-565) and was uncorrected. Mass spectroscopy was carried out on Electrospray ionization (ESI) instruments. Additionally, high-resolution mass spectra (m/z) of selected compounds were analyzed on ESI at National Development Institute of Korean Medicine (Jangheung-Gun, Jeollanam-do, Korea). All compounds for biological testing were ≥95% pure as determined by HPLC. The determination of purity was performed on a Shimadzu SCL-10A VP HPLC system using a Shimadzu Shim-pack C18 analytical column (250 mm × 4.6 mm, 5 μm, 100 Å) in two solvent systems. Solvent systems were 0.1% Formic acid in H_2_O: CH_3_CN = 90:10 to 0:100 over 30 min at a flow rate of 1 mL/min, unless noted or compound **12**, H_2_O: CH_3_CN = 95:5 to 5:95 over 30 min at a flow rate of 1 mL/min or compound **17**, H_2_O: CH_3_CN = 80:20 to 20:80 over 30 min at a flow rate of 1 mL/min. Peaks were detected by UV absorption using a diode array detector.

#### 3.1.1. General Procedure A. Synthesis of Compounds **7**, **9a**–**e**, **15** and **21**–**22**

To a mixture of 3,11-dichloro-6-methyl-6,11-dihydrodibenzo[c,f][1,2]thiazepine 5,5-dioxide (1.0 equiv) and various primary amine (1.2 equiv) in anhydrous dichloromethane, triethylamine (1.2 equiv) was added by dropwise addition. After the reaction mixture was stirred for 18 h at room temperature, it was diluted with saturated aq. NaHCO_3_, and then the organic layer was separated. The aqueous layer was further extracted with dichloromethane. The combined organic extracts were dried with anhydrous Na_2_SO_4_, filtered, and evaporated. The residue was purified by silica gel column chromatography in *n*-hexanes/ethyl acetate = 5:1 to afford **7**, **9a**–**e**, **15** and **21**–**22**.

#### 3.1.2. General Procedure B. Synthesis of Compounds **10a**–**e** and **25**

A solution of **9a**–**e, 24** in aqueous 5% NaOH was stirred vigorously for 6 h. After completion of reaction, 1N HCl solution was added to the mixture for neutralization. The reaction mixture was concentrated in vacuum, and then the residue was purified by silica gel column chromatography in chloroform/methanol = 20:1 to obtain **10a**–**e** and **25**.

#### 3.1.3. General Procedure C. Synthesis of Compounds **11a**–**e**, **12**, **13**, **14** and **24**

To a mixture of various carboxylic acid (**10a**–**e, tianeptine;** 1.0 equiv) and primary amine (serotonine, 5-Methoxytryptamine, tryptamine, **23**; 1.2 equiv) in DMF was treated with EDC∙HCl (1.2 equiv). Triethylamine (1.2 equiv) was added to the mixture by dropwise addition. After the reaction mixture was stirred for 12 h at room temperature, this mixture was partitioned ethyl acetate and saturated aq. NaHCO_3_. The aqueous layer was extracted with ethyl acetate. The combined organic extracts were dried with anhydrous Na_2_SO_4_, filtered, and evaporated. The residue was purified by silica gel column chromatography in chloroform/methanol = 30:1 to afford **11a**–**e**, **12**, **13**, **14** and **24**.

3-chloro-11-((2-(5-hydroxy-1H-indol-3-yl)ethyl)amino)-6-methyl-6,11-dihydrodibenzo[c,f][1,2]thiazepine 5,5-dioxide (**7**).

Following the general procedure A, **7** was synthesized from serotonin∙HCl (49.0 mg, 0.23 mmol). Yield 52.1 mg (74.1%); ^1^H NMR (CDCl_3_, 400 MHz) δ (ppm); 7.88 (d, *J* = 2.0 Hz, 1H), 7.36–7.39 (m, 1H), 7.29–7.35 (m, 4H), 7.24–7.28 (m, 1H), 7.18–7.21 (m, 1H), 6.95 (d, *J* = 2.4 Hz, 1H), 6.86 (d, *J* = 2.4 Hz, 1H), 6.73–6.77 (m, 1H), 4.99 (s, 1H), 3.15 (s, 3H), 2.88–2.94 (m, 2H), 2.79–2.85 (m, 2H); MS (ESI): [M + H] = 468.1.

Ethyl (3-chloro-6-methyl-5,5-dioxido-6,11-dihydrodibenzo[c,f][1,2]thiazepin-11-yl)glycinate (**9a**).

Following the general procedure A, **9a** was synthesized from glycine ethyl ester∙HCl (64.0 mg, 0.46 mmol). Yield 41.0 mg (56.7%); ^1^H NMR (CD_3_OD, 400 MHz) δ (ppm); 7.95–7.97 (m, 1H), 7.45–7.49 (m, 1H), 7.41–7.44 (m, 1H), 7.35–7.40 (m, 3H), 7.27–7.32 (m, 1H), 5.01 (s, 1H), 4.06–4.12 (m, 2H), 3.51 (s, 2H), 3.35 (s, 3H), 1.22 (t, *J* = 7.17 Hz, 3H); MS (ESI): [M + H] = 395.7.

Ethyl 3-((3-chloro-6-methyl-5,5-dioxido-6,11-dihydrodibenzo[c,f][1,2]thiazepin-11-yl)amino)propanoate (**9b**).

Following the general procedure A, **9b** was synthesized from β-alanine ethyl ester∙HCl (57.0 mg, 0.37 mmol). Yield 79.2 mg (64.6%); ^1^H NMR (CDCl_3_, 400 MHz) δ (ppm); 7.96–7.97 (m, 1H), 7.47–7.48 (m, 2H), 7.39–7.43 (m, 2H), 7.35–7.38 (m, 1H), 7.32–7.33 (m, 1H), 5.05 (s, 1H), 4.12 (q, *J* = 7.2 Hz, 2H), 3.40 (s, 3H), 2.73–2.79 (m, 2H), 2.47–2.55 (m, 2H), 1.25 (t, *J* = 7.2 Hz, 3H); MS (ESI): [M + H] = 409.8.

Ethyl 4-((3-chloro-6-methyl-5,5-dioxido-6,11-dihydrodibenzo[c,f][1,2]thiazepin-11-yl)amino)butanoate (**9c**).

Following the general procedure A, **9c** was synthesized from ethyl 4-aminobutyrate∙HCl (12.0 mg, 0.072 mmol). Yield 14.0 mg (54.3%); ^1^H NMR (CDCl_3_, 400 MHz) δ (ppm); 7.96 (d, *J* = 2.0 Hz, 1H), 7.45–7.49 (m, 1H), 7.43 (s, 1H), 7.35–7.42 (m, 3H), 7.27–7.32 (m, 1H), 5.01 (s, 1H), 4.10 (q, *J* = 7.2 Hz, 2H), 3.35 (s, 3H), 2.51 (t, *J* = 6.4 Hz, 2H), 2.32–2.38 (m, 2H), 1.77–1.84 (m, 2H), 1.22 (t, *J* = 7.2 Hz, 3H); ESI [M + H] = 423.8.

Ethyl 5-((3-chloro-6-methyl-5,5-dioxido-6,11-dihydrodibenzo[c,f][1,2]thiazepin-11-yl)amino)pentanoate (**9d**). 

Following the general procedure A, **9d** was synthesized from ethyl 5-aminopentanoate∙HCl (83.0 mg, 0.46 mmol). Yield 52.6 mg (40.1%); ^1^H NMR (CD_3_OD, 400 MHz) δ (ppm); 7.83–7.87 (m, 1H), 7.57–7.63 (m, 2H), 7.46 (dd, *J* = 7.6, 1.6 Hz, 1H), 7.36–7.42 (m, 2H), 7.28–7.34 (m, 1H), 5.06 (s, 1H), 4.08 (q, *J* = 7.2 Hz, 2H), 3.31 (s, 3H), 2.40–2.50 (m, 2H), 2.23 (t, *J* = 7.2 Hz, 2H), 1.45–1.60 (m, 4H), 1.19 (t, *J* = 7.2 Hz, 3H); ESI [M + H] = 438.8.

Ethyl 6-((3-chloro-6-methyl-5,5-dioxido-6,11-dihydrodibenzo[c,f][1,2]thiazepin-11-yl)amino)hexanoate (**9e**). 

Following the general procedure A, **9e** was synthesized from ethyl 6-aminohexanoate (73.0 mg, 0.46 mmol). Yield 89.2 mg (65.9%); ^1^H NMR (CD_3_OD, 400 MHz) δ (ppm); 7.85–7.88 (m, 1H), 7.60–7.62 (m, 2H), 7.47–7.51 (m, 1H), 7.38–7.47 (m, 2H), 7.31–7.36 (m, 1H), 5.07 (s, 1H), 4.10 (q, *J* = 7.2 Hz, 2H), 3.31 (s, 3H), 2.38–2.53 (m, 2H), 2.27 (t, *J* = 7.6 Hz, 2H), 1.45–1.59 (m, 4H), 1.25-1.34 (m, 2H), 1.22 (t, *J* = 7.2 Hz, 3H); ESI [M + H] = 451.1.

(3-chloro-6-methyl-5,5-dioxido-6,11-dihydrodibenzo[c,f][1,2]thiazepin-11-yl)glycine (**10a**).

Following the general procedure B, **10a** was synthesized from **9a** (70.0 mg, 0.18 mmol). Yield 45.2 mg (69.5%); ^1^H NMR (CD_3_OD, 400 MHz) δ (ppm); 7.97 (br, 1H), 7.72–7.75 (m, 2H), 7.62–7.64 (m, 1H), 7.51–7.54 (m, 2H), 7.40–7.45 (m, 1H), 5.59 (s, 1H), 3.34 (s, 3H), 3.28–3.31 (m, 2H); MS (ESI): [M + H] = 367.0.

3-((3-chloro-6-methyl-5,5-dioxido-6,11-dihydrodibenzo[c,f][1,2]thiazepin-11-yl)amino)propanoic acid (**10b**).

Following the general procedure B, **10b** was synthesized from **9b** (69.7 mg, 0.17 mmol). Yield 55.8 mg (86.2%); ^1^H NMR (CD_3_OD, 400 MHz) δ (ppm); 7.93 (d, *J* = 2.4 Hz, 1H), 7.65–7.73 (m, 2H), 7.49–7.57 (m, 3H), 7.36–7.42 (m, 1H), 5.34 (s, 1H), 3.36 (s, 3H), 2.78–2.89 (m, 2H), 2.46 (t, *J*=6.4 Hz, 2H); MS (ESI): [M + H] = 381.8.

4-((3-chloro-6-methyl-5,5-dioxido-6,11-dihydrodibenzo[c,f][1,2]thiazepin-11-yl)amino)butanoic acid (**10c**).

Following the general procedure B, **10c** was synthesized from **9c** (120.0 mg, 0.28 mmol). Yield 51.0 mg (45.5%); ^1^H NMR (CD_3_OD, 400 MHz) δ (ppm); 7.89–7.93 (m, 1H), 7.64–7.71 (m, 2H), 7.52–7.56 (m, 1H), 7.44–7.51 (m, 2H), 7.35–7.40 (m, 1H), 5.30 (s, 1H), 3.34 (s, 3H), 2.54–2.64 (m, 2H), 2.29–2.35 (m, 2H), 1.74–1.85 (m, 2H); MS (ESI): [M + H] = 395.8.

5-((3-chloro-6-methyl-5,5-dioxido-6,11-dihydrodibenzo[c,f][1,2]thiazepin-11-yl)amino)pentanoic acid (**10d**).

Following the general procedure B, **10d** was synthesized from **9d** (100.7 mg, 0.23 mmol). Yield 86.9 mg (92.2%); ^1^H NMR (CDCl_3_, 400 MHz) δ (ppm); 7.96–7.98 (m, 1H), 7.41–7.50 (m, 3H), 7.34–7.39 (m, 2H), 7.27–7.31 (m, 1H), 5.08 (s, 1H), 3.29 (s, 3H), 2.36–2.54 (m, 2H), 2.25–2.32 (m, 2H), 1.46–1.64 (m, 4H); MS (ESI): [M + H] = 408.8.

6-((3-chloro-6-methyl-5,5-dioxido-6,11-dihydrodibenzo[c,f][1,2]thiazepin-11-yl)amino)hexanoic acid (**10e**).

Following the general procedure B, **10e** was synthesized from **9e** (89.0 mg, 0.20 mmol). Yield 77.3 mg (92.7%); ^1^H NMR (CD_3_OD, 400 MHz) δ (ppm); 7.89–7.91 (m, 1H), 7.64–7.66 (m, 2H), 7.49–7.52 (m, 1H), 7.41–7.48(m, 2H), 7.33–7.39(m, 1H), 5.18 (s, 1H), 3.32 (s, 3H), 2.45–2.56(m, 2H), 2.23 (t, *J* = 7.2 Hz, 2H), 1.46–1.60 (m, 4H), 1.28–1.36 (m, 2H); MS (ESI): [M + H] = 423.9.

2-((3-chloro-6-methyl-5,5-dioxido-6,11-dihydrodibenzo[c,f][1,2]thiazepin-11-yl)amino)-*N*-(2-(5-hydroxy-1H-indol-3-yl)ethyl)acetamide (**11a**).

Following the general procedure C, **11a** was synthesized from **10a** (55.0 mg, 0.15 mmol). Yield 65.5 mg (83.2%); ^1^H NMR (CDCl_3_, 400 MHz) δ (ppm); 8.22–8.24 (m, 1H), 7.92 (d, *J* = 2.0 Hz, 1H), 7.36 (dd, *J* = 8.2, 2.1 Hz, 1H), 7.28–7.31 (m, 1H), 7.18–7.25 (m, 2H), 7.12–7.13 (m, 1H), 7.01–7.06 (m, 1H), 6.96–6.98 (m, 1H), 6.84 (dd, *J* = 8.8, 2.4 Hz, 1H), 6.72–6.76 (m, 1H), 4.61 (s, 1H), 3.69–3.77 (m, 1H), 3.41–3.50 (m, 1H), 3.11 (s, 3H), 2.84–3.02 (m, 2H), 1.89–2.05 (m, 2H); ^13^C NMR (CDCl_3_, 100 MHz): δ 171.4, 151.0, 139.1, 139.1 135.0, 134.4, 132.7, 132.7, 131.2, 131.2, 129.4, 128.5, 128.5, 127.7, 127.2, 122.8, 112.6, 112.4, 111.8, 103.0, 77.2, 50.0, 40.0, 39.2, 25.2; MS (ESI): [M + H] = 524.9.

3-((3-chloro-6-methyl-5,5-dioxido-6,11-dihydrodibenzo[c,f][1,2]thiazepin-11-yl)amino)-*N*-(2-(5-hydroxy-1H-indol-3-yl)ethyl)propanamide (**11b**).

Following the general procedure C, **11b** was synthesized from **10b** (36.0 mg, 0.09 mmol). Yield 47.1 mg (92.4%); ^1^H NMR (CDCl_3_, 400 MHz) δ (ppm); 8.13-8.14 (m, 1H), 7.91 (d, *J* = 2.0 Hz, 1H), 7.40 (dd, *J* = 8.2, 2.1 Hz, 1H), 7.24–7.28 (m, 1H), 7.17–7.20 (m, 1H), 7.08–7.13 (m, 2H), 7.02 (d, *J* = 2.1 Hz, 1H), 6.77–6.85 (m, 3H), 4.51 (s, 1H), 3.56–3.63 (m, 1H), 3.24–3.34 (m, 1H), 3.11 (s, 3H), 2.74–2.90 (m, 2H), 2.45–2.65 (m, 2H), 2.16–2.32 (m, 2H); ^13^C NMR (DMSO-*d*_6_, 100 MHz): δ 171.0, 150.2, 141.0, 137.9, 137.5, 132.9, 132.1, 132.1, 132.0, 130.8, 129.3, 129.3, 128.6, 128.3, 127.9, 126.7, 123.0, 111.7, 111.3, 110.9, 102.2, 79.2, 45.9, 44.0, 38.3, 35.7, 25.4; MS (ESI): [M + H] = 538.9.

4-((3-chloro-6-methyl-5,5-dioxido-6,11-dihydrodibenzo[c,f][1,2]thiazepin-11-yl)amino)-*N*-(2-(5-hydroxy-1H-indol-3-yl)ethyl)butanamide (**11c**).

Following the general procedure C, **11c** was synthesized from **10c** (45.0 mg, 0.11 mmol). Yield 56.8 mg (90.2%); ^1^H NMR (CD_3_OD, 400 MHz) δ (ppm); 7.86–7.89 (m, 1H), 7.58–7.61 (m, 2H), 7.30–7.48 (m, 4H), 7.14 (d, *J* = 8.8 Hz, 1H), 6.99 (s, 1H), 6.92 (d, *J* = 2.4 Hz, 1H), 6.64 (dd, *J* = 8.7, 2.2 Hz, 1H), 5.06 (s, 1H), 3.41-3.46 (m, 2H), 3.34–3.36 (m, 3H), 2.82-2.88 (m, 2H), 2.35–2.49 (m, 2H), 2.11 (t, *J* = 7.4 Hz, 2H), 1.40–1.57 (m, 2H); MS (ESI): [M + H] = 554.0.

5-((3-chloro-6-methyl-5,5-dioxido-6,11-dihydrodibenzo[c,f][1,2]thiazepin-11-yl)amino)-*N*-(2-(5-hydroxy-1H-indol-3-yl)ethyl)pentanamide (**11d**).

Following the general procedure C, **11d** was synthesized from **10d** (50.0 mg, 0.12 mmol). Yield 61.0 mg (88.0%); ^1^H NMR (CD_3_OD, 400 MHz) δ (ppm); 7.86–7.89 (m, 1H), 7.58–7.61 (m, 2H), 7.30–7.48 (m, 4H), 7.14 (d, *J* = 8.8 Hz, 1H), 6.99 (s, 1H), 6.92 (d, *J* = 2.4 Hz, 1H), 6.64 (dd, *J* = 8.7, 2.2 Hz, 1H), 5.06 (s, 1H), 3.41–3.46 (m, 2H), 3.34–3.36 (m, 3H), 2.82–2.88 (m, 2H), 2.35–2.49 (m, 2H), 2.11 (t, *J* = 7.4 Hz, 2H), 1.40–1.57 (m, 4H); MS (ESI): [M + H] = 568.1.

6-((3-chloro-6-methyl-5,5-dioxido-6,11-dihydrodibenzo[c,f][1,2]thiazepin-11-yl)amino)-*N*-(2-(5-hydroxy-1H-indol-3-yl)ethyl)hexanamide (**11e**).

Following the general procedure C, **11e** was synthesized from **10e** (75.0 mg, 0.17 mmol). Yield 69.0 mg (67.0%); ^1^H NMR (CDCl_3_, 400 MHz) δ (ppm); 8.07 (br, 1H), 7.92–7.95 (m, 1H), 7.36–7.45 (m, 3H), 7.30–7.34 (m, 2H), 7.13 (d, *J* = 8.8 Hz, 1H), 6.88–6.90 (m, 2H), 6.72 (dd, *J* = 8.5, 2.1 Hz, 1H), 4.95 (s, 1H), 3.46–3.50 (m, 2H), 3.30 (s, 3H), 2.80–2.84 (m, 2H), 2.36–2.45 (m, 2H), 1.93–2.04 (m, 2H), 1.40–1.52 (m, 4H), 1.21–1.26 (m, 2H); ^13^C NMR (CDCl_3_, 100 MHz): δ 173.5, 150.1, 139.8, 138.6, 137.4, 136.4, 134.2, 132.4, 131.9, 131.2, 129.4, 128.3, 128.1, 128.1, 127.2, 123.1, 112.2, 111.9, 111.9, 111.9, 103.0, 77.2, 47.6, 47.2, 40.0, 38.9, 26.3, 26.6, 26.2, 25.3; ESI [M + H] = 582.8.

7-((3-chloro-6-methyl-5,5-dioxido-6,11-dihydrodibenzo[c,f][1,2]thiazepin-11-yl)amino)-*N*-(2-(5-hydroxy-1H-indol-3-yl)ethyl)heptanamide (**12**).

Following the general procedure C, **12** was synthesized from tianeptine (12.0 mg, 0.028 mmol) and serotonin∙HCl (7.2 mg, 0.034 mmol). Yield 10.5 mg (63.0%); 75%; m.p.: 92–95 ℃; ^1^H NMR (CD_3_OD, 400 MHz) δ (ppm); 7.87 (s, 1H), 7.58–7.61 (m, 2H), 7.30–7.48 (m, 4H), 7.14 (d, *J* = 9.2 Hz, 1H), 6.99 (s, 1H), 6.92 (d, *J* = 2.4 Hz, 1H), 6.65 (dd, *J* = 8.8, 2.4 Hz, 1H), 5.06 (s, 1H), 3.43 (t, *J* = 7.2 Hz, 2H), 3.34 (s, 3H), 2.84 (t, *J* = 6.8 Hz, 2H), 2.35–2.49 (m, 2H), 2.11 (t, *J* = 7.6 Hz, 2H), 1.40–1.57(m, 4H), 1.17–1.31(m, 4H); ^13^C NMR (DMSO-*d*_6_, 100 MHz): δ 173.3, 150.0, 139.9, 138.8, 136.3, 134.4, 132.4, 132.4, 132.0, 131.3, 129.5, 128.5, 128.5, 128.0, 127.7, 123.0, 112.4, 112.2, 111.8, 111.9, 103.2, 77.2, 47.8, 40.0, 38.9, 36.6, 29.4, 28.8, 26.7, 25.3, 25.2; ESI [M + H] = 596.6; HRMS (ESI) [M + H]^+^ (C_31_H_35_ClN_4_O_4_S): calcd. 595.2140, found. 595.2086.

7-((3-chloro-6-methyl-5,5-dioxido-6,11-dihydrodibenzo[c,f][1,2]thiazepin-11-yl)amino)-*N*-(2-(5-methoxy-1H-indol-3-yl)ethyl)heptanamide (**13**).

Following the general procedure C, **13** was synthesized from tianeptine (20.0 mg, 0.046 mmol) and 5-methoxytryptamine∙HCl (12.0 mg, 0.052 mmol). Yield 21.7 mg (77.4%); ^1^H NMR (CD_3_OD, 400 MHz) δ (ppm); 7.87 (s, 1H), 7.57–7.61 (m, 2H), 7.27–7.47 (m, 4H), 7.18 (d, *J* = 8.8 Hz, 1H), 6.98–7.06 (m, 2H), 6.70 (dd, *J* = 8.8, 2.4 Hz, 1H), 5.06 (s, 1H), 3.81 (s, 3H), 3.41–3.47 (m, 2H), 3.34 (s, 3H), 2.88 (t, *J* = 7.6 Hz, 2H), 2.32–2.47 (m, 2H), 2.06–2.13 (t, *J* = 7.6 Hz, 2H), 1.40–1.54 (m, 4H), 1.16–1.28 (m, 4H); ESI [M + H] = 610.5; HRMS (ESI) [M + H]^+^ (C_32_H_37_ClN_4_O_4_S): calcd. 609.2297, found. 609.2266.

*N*-(2-(1H-indol-3-yl)ethyl)-7-((3-chloro-6-methyl-5,5-dioxido-6,11-dihydrodibenzo[c,f][1,2]thiazepin-11-yl)amino)heptanamide (**14**).

Following the general procedure C, **14** was synthesized from tianeptine (48.0 mg, 0.11 mmol) and tryptamine∙HCl (26.0 mg, 0.13 mmol). Yield 42.3 mg (66.4%); ^1^H NMR (DMSO-*d*_6_, 400 MHz) δ (ppm); 7.91 (s, 1H), 7.78–7.85 (m, 1H), 7.73 (s, 1H), 7.65–7.70 (m, 2H), 7.43–7.48 (m, 2H), 7.33-7.36 (m, 1H), 7.29 (t, *J* = 8.0 Hz, 1H), 7.08 (s, 1H), 7.00 (t, *J* = 8.0 Hz, 1H), 6.92 (t, *J* = 7.6 Hz, 1H), 5.12 (s, 1H), 3.32–3.34 (m, 2H), 3.30 (s, 3H), 2.74 (t, *J* = 7.2 Hz, 2H), 2.30–2.37 (m, 2H), 1.97 (t, *J*=7.2 Hz, 2H), 1.32–1.45 (m, 4H), 1.10–1.23 (m, 4H): ESI [M + H]=580.7; HRMS (ESI) [M+H]^+^ (C_31_H_35_ClN_4_O_3_S): calcd. 597.2191, found. 597.2150.

3-chloro-11-((7-hydroxyheptyl)amino)-6-methyl-6,11-dihydrodibenzo[c,f][1,2]thiazepine 5,5-dioxide (**15**).

Following the general procedure A, **15** was synthesized from 7-Aminoheptanol (73.1 mg, 0.36 mmol). Yield 41.0 mg (56.7%); ^1^H NMR (CD_3_OD, 400 MHz) δ (ppm); 7.85 (s, 1H), 7.58–7.60 (m, 2H), 7.43–7.47 (m, 1H), 7.37–7.42 (m, 2H), 7.27–7.34 (m, 1H), 5.05 (s, 1H), 3.48 (t, *J* = 6.8 Hz, 2H), 3.29 (s, 3H), 2.35-2.51 (m, 2H), 1.40–1.52 (m, 4H), 1.19–1.34 (m, 6H); MS (ESI): [M + H] = 423.8.

7-((3-chloro-6-methyl-5,5-dioxido-6,11-dihydrodibenzo[c,f][1,2]thiazepin-11-yl)amino)heptanal (**16**).

A solution of oxalyl chloride (45.0 mg, 0.35 mmol) in dry dichloromethane was prepared under N_2_ atmosphere at −8 ℃. After the solution was stirred for 10 min, dry DMSO (50.0 μL, 0.71 mmol) was rapidly added. The reaction mixture was stirred for 20 min at −78 ℃, then a solution of **15** (50.0 mg, 0.12 mmol) in dry dichloromethane was added. After 1 h, dry triethylamine (98.0 μL, 0.71 mmol) was added, and the reaction mixture was warmed to 0 ℃ and stirred for 1hr. After completion of reaction, the mixture was diluted with saturated aq. NH_4_Cl and extracted with dichloromethane. The combined organic extracts were dried with anhydrous Na_2_SO_4_, filtered, and evaporated. The residue was purified by silica gel column chromatography in *n*-hexanes/ethyl acetate = 2:1 to afford **16.** Yield 35.0 mg (70.3%); ^1^H NMR (CDCl_3_, 400 MHz) δ (ppm); 9.72–9.74 (m, 1H), 7.93–7.96 (d, *J* = 2.0 Hz, 1H), 7.44–7.46 (d, *J* = 2.0 Hz, 1H), 7.42 (s, 1H), 7.38–7.41 (m, 1H), 7.36–7.37 (m, 2H), 7.27–7.31 (m, 1H), 4.97 (s, 1H), 3.34 (s, 3H), 2.35–2.47 (m, 4H), 1.55–1.65 (m, 2H), 1.40–1.52 (m, 2H), 1.20–1.34 (m, 4H); MS (ESI): [M + H] = 421.9.

3-chloro-11-((7-((2-(5-hydroxy-1H-indol-3-yl)ethyl)amino)heptyl)amino)-6-methyl-6,11-dihydrodibenzo[c,f][1,2]thiazepine 5,5-dioxide (**17**).

To a mixture of **16** (12.0 mg, 0.029 mmol) and serotonin∙HCl (8.1 mg, 0.038 mmol) in a solution of 1,2-dichloroethane/methanol (5:1, *v*/*v*), acetic acid (2.0 μL, 0.038 mmol) was added. After 5 min, NaCNBH_3_ (4.0 mg, 0.058 mmol) was added and the reaction mixture was stirred at room temperature for 1 h under an atmosphere of Ar. The reaction mixture was diluted with saturated aq. NaHCO_3_ and extracted with ethyl acetate. The combined organic extracts were dried with anhydrous Na_2_SO_4_, filtered, and evaporated. The residue was purified by silica gel column chromatography in chloroform/methanol = 20:1 to afford **17**. Yield 10.3 mg (61.1%); ^1^H NMR (CD_3_OD, 400 MHz) δ (ppm); 7.86 (s, 1H), 7.58–7.61 (m, 2H), 7.46 (d, *J* = 7.6 Hz, 1H), 7.37–7.42 (m, 2H), 7.29–7.34 (m, 1H), 7.15 (d, *J* = 8.8 Hz, 1H), 7.02 (s, 1H), 6.88–6.90 (m, 1H), 6.63–6.68 (m, 1H), 5.03 (s, 1H), 3.26 (s, 3H), 2.92–3.03 (m, 4H), 2.68–2.74 (m, 2H), 2.34–2.48 (m, 2H), 1.35–1.55 (m, 4H), 1.15–1.28 (m, 6H); MS (ESI): [M + H] = 581.9; HRMS (ESI) [M − H]^−^ (C_31_H_37_ClN_4_O_3_S): calcd. 579.2202, found. 579.2173.

Methyl (3S)-3-((7-((3-chloro-6-methyl-5,5-dioxido-6,11-dihydrodibenzo[c,f][1,2]thiazepin-11-yl)amino)heptanamido)methyl)-5-methylhexanoate (**24**).

Following the general procedure C, **24** was synthesized from tianeptine (35.0 mg, 0.080 mmol) and **23** (16.6 mg, 0.096 mmol). Yield 43.5 mg (91.7%); ^1^H NMR(CDCl_3_, 400 MHz) δ (ppm); 7.95 (d, *J* = 2.1 Hz, 1H), 7.43–7.49 (m, 2H), 7.33–7.41 (m, 3H), 7.27–7.32 (m, 1H), 4.99 (s, 1H), 3.66 (s, 3H), 3.36 (s, 3H), 3.27–3.34 (m, 1H), 3.08–3.17 (m, 1H), 2.43–2.48 (m, 2H), 2.21–2.35 (m, 2H), 2.09–2.16 (m, 2H), 1.22–1.70 (m, 10H), 1.08–1.20 (m, 2H), 0.82–0.93 (m, 6H); MS (ESI): [M + H] = 591.9.

(3S)-3-((7-((3-chloro-6-methyl-5,5-dioxido-6,11-dihydrodibenzo[c,f][1,2]thiazepin-11-yl)amino)heptanamido)methyl)-5-methylhexanoic acid (**25**).

Following the general procedure B, **25** was synthesized from **24** (30.0 mg, 0.05 mmol). Yield 25.1 mg (85.7%); ^1^H NMR (CDCl_3_, 400 MHz) δ (ppm); 7.99 (br, 1H), 7.43–7.56 (m, 3H), 7.28–7.43 (m, 3H), 5.16–5.17 (m, 1H), 3.26–3.28 (m, 3H), 3.24–3.31 (m, 1H), 3.05–3.15 (m, 1H), 2.26–2.60 (m, 3H), 2.06–2.18 (m, 3H), 1.41–1.71 (m, 5H), 1.10–1.35 (m, 7H), 0.85–0.95 (m, 6H); ^13^C NMR (DMSO-*d*_6_, 100 MHz): δ 177.5, 173.7, 139.9, 139.4, 135.3, 133.1, 133.1, 132.8, 132.8, 132.0, 130.3, 128.5, 128.5, 127.4, 77.2, 46.7, 44.7, 42.3, 39.4, 39.1, 36.1, 32.5, 28.0, 27.8, 25.4, 25.1, 24.9, 22.7, 22.6; MS (ESI): [M − H] = 576.0.

3-chloro-6-methyl-11-((7-(methyl(3-phenyl-3-(4-(trifluoromethyl)phenoxy)propyl)amino)heptyl)amino)-6,11-dihydrodibenzo[c,f][1,2]thiazepine 5,5-dioxide (**26**).

Following the procedure for the synthesis of **17**, **26** was synthesized from **16** (15.0 mg, 0.036 mmol). Yield 19.8 mg (77.8%); ^1^H NMR (CDCl_3_, 400 MHz) δ (ppm); 7.97 (d, *J* = 2.0 Hz, 1H), 7.45–7.48 (m, 1H), 7.36–7.44 (m, 6H), 7.33–7.35 (m, 4H), 7.28–7.32 (m, 1H), 7.27 (s, 1H), 6.89–6.92 (m, 2H), 5.28–5.31 (m, 1H), 5.00 (s, 1H), 3.38 (s, 3H), 2.51–2.58 (m, 1H), 2.41–2.46 (m, 3H), 2.27–2.31 (m, 2H), 2.21 (s, 3H), 2.11–2.17 (m, 1H), 1.92-2.01 (m, 1H), 1.36–1.49 (m, 4H), 1.16–1.27 (m, 6H); MS (ESI): [M + H] = 714.2.

3-(2-((7-((3-chloro-6-methyl-5,5-dioxido-6,11-dihydrodibenzo[c,f][1,2]thiazepin-11-yl)amino)heptyl)amino)ethyl)-1H-indole-5-carbonitrile (**27**).

Following the procedure for the synthesis of **17**, **27** was synthesized from **16** (15.0 mg, 0.036 mmol). Yield 14.2 mg (67.5%); ^1^H NMR (CDCl_3_, 400 MHz) δ (ppm); 7.97 (d, *J* = 2.0 Hz, 1H), 7.91 (s, 1H), 7.47–7.50 (m, 1H), 7.40-7.44 (m, 2H), 7.29–7.36 (m, 5H), 7.21–7.24 (m, 1H), 4.99 (s, 1H), 3.32 (s, 3H), 3.12–3.22 (m, 4H), 2.82–2.86 (m, 2H), 2.32–2.45 (m, 2H), 1.99–2.05 (m, 2H), 1.64–1.68 (m, 2H), 1.16–1.28 (m, 6H); MS (ESI): [M + H] = 590.1.

### 3.2. Neurotransmitter Reuptake Assay and Data Analysis

DA, NE, and serotonin neurotransmitter uptake activities were measured using the Neurotransmitter Transporter Uptake Assay Kit (Molecular Devices, Sunnyvale, CA, USA) according to the manufacturer’s instructions. Cells were seeded in 50 μg/mL, poly-L-lysine coated, 96-well, black-walled, clear-bottomed plates (Corning, NY, USA) at a density of 5 × 10^4^ cells/well and cultured overnight. The prepared cells were washed three times with the HEPES-buffered solution (150 mM NaCl, 5 mM KCl, 10 mM Glucose, 2 mM CaCl_2_, 1 mM MgCl_2_, 10 mM HEPES, and pH 7.4) using an EL×405 Select plate washer (BioTek Instruments, Winooski, VT, USA). The assay loading dye solution was added to the cells and the fluorescent intensity was measured for 30 min. The cells were excited at 440 nm light and the emission was collected every 10 s at 520 nm light. Cells treated with test compounds were incubated for 15 min in a humidified atmosphere of 5% CO_2_ at 37 °C and the assay loading dye solution was then added to the cells. The fluorescent intensity was measured for 30 min using the FDSS6000 96-well fluorescence plate reader (Hamamatsu Photonics, Hamamatsu, Japan). In order to acquire the dose-response curve, the percent inhibition was calculated using the endpoint measurements at 30 min of test compound-treated and untreated control cells. IC_50_ values for inhibition of transporter activity were determined using a four parameter logistic non-linear regression model with the GraphPad Prism6 software (GraphPad Software, La Jolla, CA, USA). 

### 3.3. Intracellular Calcium Mobilization Measurement by Using FLIPR Tetra System

Intracellular calcium mobilization was measured by using FLIPR Tetra System (Molecular Devices, Sunnyvale, CA, USA) in accordance with manufacturer’s instruction. Briefly, HEK-hKOR and HEK-hMOR (provided by Professor Ki Duk Park, Korea Institute of Science and Technology) were seeded in 96-well clear-bottom black plate in 100 μL culture medium at a density of 30,000 cells/well and then incubated in a 5% CO_2_, 37 °C incubator overnight. After incubation, cells were added with 100 μL of the calcium dye dissolved in loading buffer (1 × HBSS plus 20 mM HEPES buffer) for 2 h at 37 °C. Subsequently, the plate was transferred to the FLIPR tetra system for treating 50 μL of the 5× compounds diluted in loading buffer and reading intracellular calcium mobilization. The final concentration of DMSO was 0.1%. Fluorescence was measured at 1 s intervals for 60 s by using a cooled CCD camera with 470–495 nm excitation LED module and 515–575 nm emission filter and fluorescence signal of calcium mobilization was used as the maximum value minus minimum. A dose response curve was generated with the changes in Relative Fluorescence Units (∆RFU) versus Log_10_ dose.

### 3.4. β-Arrestin Recruitment Assay at hOPRM1 Receptor

PathHunter^®^ CHO-K1 OPRM1 β-arrestin recruitment assay (93-0213, DiscoveRx, Fremont, CA, USA) according to the manufacturer’s instructions. The assay was designed to measure β-arrestin recruitment to the OPRM1 GPCR using a principal enzyme fragment complementation (EFC) technology. This EFC technology utilized a genetically engineered β-galactosidase enzyme consisting of two fragments—a large protein fragment (Enzyme Acceptor, EA, tagged β-arrestin) and a small peptide fragment (Enzyme Donor, ED, tagged GPCR). The resulting functional enzyme hydrolyzes substrate to generate a chemiluminescent signal is detected with a microplate reader (SpectraMax^®^i3 multi-mode microplate reader; Molecular Device, Sunnyvale, CA, USA). A dose response curve was generated with the changes in Relative Luminescence Units (∆RLU) versus Log_10_ dose.

### 3.5. In Vivo Efficacy

Adult male Sprague-Dawley rats weighing 150–180 g each were used. The animals were housed in a room with a constant temperature of 22–23 °C and an alternating 12-h light/dark cycle, with unrestricted access to food and water. Implantation of intrathecal catheters was performed for drug administration under sevoflurane anesthesia, as described previously [25]. Seven days of recovery were allowed after surgery before the use of catheterized rats in the experiments. Spinal nerve-ligated neuropathic pain was induced by tightly ligating the left L5 and L6 spinal nerves, as described previously [26]. Only rats showing allodynia after either spinal nerve ligation (withdrawal threshold < 4 g) were studied. The von Frey test was used to determine the 50% probability paw withdrawal threshold (PWT), using the up and down method [27], in which a positive response was defined as brisk withdrawal or paw flinching during or immediately after application of a filament. On experimentation days (12 days after SNL), rats were acclimatized in a box with a wire mesh floor for at least 20 min before being randomly allocated to one of the experimental groups. All experiments were carried out by an observer who was blind to the treatments. To investigate the effects of **12** (30 or 100 μg) and **17** (10 or 30 μg), it was supplied intrathecally following post-injury baseline PWT determination in rats with either spinal nerve-ligated neuropathic pain. Withdrawal thresholds were determined at 15, 30, 60, 90, 120, 150, and 180 min after intrathecal administration.

### 3.6. Statistical Analysis

All data are expressed as means ± SEM. The time-response data are presented as the withdrawal threshold in grams. The dose-response data are presented as a percentage of the maximum possible effect (%MPE): %MPE = ((post-drug threshold−post-injury baseline threshold)/(cutoff threshold−post-injury baseline threshold)) × 100. Time and dose-response data were analyzed by a one-way and two-way repeated measures analysis of variance (ANOVA), with a Bonferroni’s post hoc test. A *p*-value < 0.05 was considered statistically significant.

## 4. Conclusions

In this article, we reported the development of multifunctional compounds for the treatment of neuropathic pain. The key pharmacophores of currently used clinical pain drugs, including pregabalin, fluoxetine and serotonin analogs, were hybridized to the side chain of tianeptine, which has been used as an antidepressant. The biological activities of the hybrid analogs were evaluated at human transporters of neurotransmitters, including serotonine (hSERT), norepinephrine (hNET) and dopamine (hDAT), as well as mu (μ) and kappa (κ) opioid receptors. A representative compound, **17**, exhibited multiple transporter inhibitory activities for the uptake of neurotransmitters with IC_50_ values of 70 nM, 154 nM and 2.01 μM at hSERT, hNET and hDAT, respectively. In addition, compound **17** showed partial agonism (EC_50_ = 384 nM) at the m-opioid receptor, with lower levels of β-arrestin recruitment than compound **12**. In in vivo pain animal experiments, the multifunctional compound **17** showed significantly reduced allodynia in SNL by intrathecal administration, indicating that multitargeted strategies in single therapy can considerably benefit patients with multifactorial diseases, such as pain and depression. However, due to the low metabolic stability of the developed MFCs, there were limitations to determine the PK properties of MFCs and appropriate routes of drug administration for clinical application. Therefore, lead optimization research with the aim of improving pharmacokinetic properties should be conducted in the future based on this study, which may change the paradigm of drug development targeting multifactorial diseases.

## Data Availability

Data is contained within the article and Appendix A.

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
