# Peer review of "Development of Dibenzothiazepine Derivatives as Multifunctional Compounds for Neuropathic Pain"

_pharmaceuticals, 2022, doi:10.3390/ph15040407_

Round 1

Reviewer 1 Report

The study is reasonably well-presented and experiments conducted are of a decent standard. However, I have concerns regarding the potential in vivo effects of these compounds. Most of the study deals with chemical modification of existing drugs and only superficially examines their in vivo actions in neuropathic pain. The team should also be clearer about whether they are designing these MFCs for neuropathic pain and depression or neuropathic pain alone. Do the team aim to treat depression associated with neuropathic pain (see PMID: 34862336)? The discussion of this is lacking. In any case, further in vivo characterization of these compounds is required. My concerns are outlined below:

Major concerns

Since these compounds are derived from antidepressants, the study authors should perform experiments to address whether they are still effective as antidepressants in addition to being effective for neuropathic pain. The team should determine the effect of compounds in depression-like behavioral measures in mice such as the sucrose splash test (see PMID: 20436931).

The team state that compound 17 shows partial agonism at MORs. Therefore, the team should determine whether this compound is prone to abuse liability by performing conditioned place preference tests.

For studies involving behavioral assessment of withdrawal responses, please provide additional data indicating baseline measures of injured mice compared to naïve and/or sham controls.

Minor concerns

Figure 1A is missing statistical analyses of groups. Please indicate on the graph and the figure legend.

Figure 1B: Have statistical analyses been used to compare 30 ug v vehicle and 100 ug v 30 ug groups?

Figure 2A is missing statistical analyses of groups. Please indicate on the graph and the figure legend.

Figure 2B: Have statistical analyses been used to compare 30 ug v vehicle and 100 ug v 30 ug groups?

Author Response

The study is reasonably well-presented and experiments conducted are of a decent standard. However, I have concerns regarding the potential in vivo effects of these compounds. Most of the study deals with chemical modification of existing drugs and only superficially examines their in vivo actions in neuropathic pain. The team should also be clearer about whether they are designing these MFCs for neuropathic pain and depression or neuropathic pain alone. Do the team aim to treat depression associated with neuropathic pain (see PMID: 34862336)? The discussion of this is lacking. In any case, further in vivo characterization of these compounds is required. My concerns are outlined below:

Major concerns

Since these compounds are derived from antidepressants, the study authors should perform experiments to address whether they are still effective as antidepressants in addition to being effective for neuropathic pain. The team should determine the effect of compounds in depression-like behavioral measures in mice such as the sucrose splash test (see PMID: 20436931).

=> We would like to appreciate the valuable comments regarding the proposal of MFCs` new indication for the treatment effect of depression. The research project was started with the aim of treating neuropathic pain through the development of MFCs based on the results of previous studies that Tianeptine showed analgesic activity in pain animal models (Life Sci 1999, 64 (15), 1313-9, Pain Physician 2017, 20 (4), E593-E600, Neurosci Lett 2014, 583, 103-7, Anesth Analg 2012, 114 (3), 683-9). Also, this study aimed to verify in vivo POC for neuropathic pain through the development of novel MFCs derived from Tianeptine, which successfully showed significant efficacy for the treatment of neuropathic pain.

We admit that additional in-vivo experiment (depression-like behavioral measures in mice such as the sucrose splash test) suggested by the reviewer will be a valuable new research topic since tianeptine has been clearly demonstrated to alleviate anxious symptoms associated with depression (Mol Psychiatry 2010, 15 (3), 237-49, CNS Drugs 2008, 22 (1), 15-26) and unlike Tianeptine, the new MFCs developed in this study have shown strong serotonin reuptake inhibitory activities.

Therefore, we think that it would be better to develop lead compounds of MFCs with improved metabolic stabilities and brain permeability in the future and then proceed the studies of depression associated with neuropathic pain as suggested.

Furthermore, we added the limitations of developed MFCs at “4. Conclusions” in the main manuscript as below.

“In in vivo pain animal experiments, the multifunctional compound 17 showed significantly reduced allodynia in SNL by intrathecal administration, indicating that multi-targeted strategies in single therapy can considerably benefit patients with multifactorial diseases, such as pain and depression. However, due to the low metabolic stability of the developed MFCs, there were limitations to determine the PK properties of MFCs and appropriate routes of drug administration for clinical application. Therefore, lead optimization research with the aim of improving pharmacokinetic properties should be conducted in the future based on this study, which may change the paradigm of drug development targeting multifactorial diseases.”

The team state that compound 17 shows partial agonism at MORs. Therefore, the team should determine whether this compound is prone to abuse liability by performing conditioned place preference tests.

=> This study mainly focused on verification of in vivo POC through the development of novel MFCs, which successfully showed efficacy in the treatment of neuropathic pain. Therefore, we consider that it would be better to conduct the research on m-opioid side effects such as abuse liability of MFCs after lead optimization in the future.

For studies involving behavioral assessment of withdrawal responses, please provide additional data indicating baseline measures of injured mice compared to naïve and/or sham controls. 

=> According to the reviewer’s suggestion, we revised the figure 1 and 2.

Minor concerns

Figure 1A is missing statistical analyses of groups. Please indicate on the graph and the figure legend.

=> Figure 1A is simply a graph showing the effect of each dose over time, and the statistical significance of this result is expressed in Figure 1B by analyzing the %MPE by dose by ANOVA with the overall interpretation of Figure 1. Therefore, even the statistical analysis of 1B is sufficient to explain the overall result in Figure 1. The legend has been corrected.

Figure 1B: Have statistical analyses been used to compare 30 ug v vehicle and 100 ug v 30 ug groups?

=> Yes, we confirm that the statistical analyses were conducted as the contents in the question.

Figure 2A is missing statistical analyses of groups. Please indicate on the graph and the figure legend.

=> Please refer to the answer related to Figure 1A.

Figure 2B: Have statistical analyses been used to compare 30 ug v vehicle and 100 ug v 30 ug groups?

=> Yes, we confirm that the statistical analyses were conducted as the contents in the question.

Reviewer 2 Report

Please add the safety aspects that were checked or need to be further analyzed, for example for in vivo tests.

The ethical approval is dated November 2017. Does this mean that it is for long-term approval until now, 2022? 

Can the route of delivery of new compound be other forms in addition to intrathecal? for translation to human and easier routes of administration?

Please add the limitations of this study. 

The model used for efficacy used was withdrawal to von Frey. Please elaborate if other tests might be required in addition to mechanical sensitivity.

Author Response

Please add the safety aspects that were checked or need to be further analyzed, for example for in vivo tests.

=> We described the safety aspects as below at 2.2.3. In Vivo Behavioral Responses in the main manuscript.

“To evaluate the behavioral changes induced by compounds 12 and 17, additional rats were examined for 3 hrs after intrathecal administration of the highest doses of the agents used. The motor function and pinna and corneal reflexes were normal after intrathecal administration of compounds 12 and 17 at the highest doses. Overt abnormal behavioral changes were not observed.”

The ethical approval is dated November 2017. Does this mean that it is for long-term approval until now, 2022? 

=> Yes, it is correct.

Can the route of delivery of new compound be other forms in addition to intrathecal? for translation to human and easier routes of administration?

=> The in vitro metabolic stabilities of compounds 12 and 17 were determined to be extremely low (Table S1, with percent remaining after 30 min incubation with the S9 microsomal fraction). For this reason, we performed in vivo experiments at the spinal level by intrathecal administration only.

Therefore, we think that it would be better to develop lead compounds of MFCs with improved metabolic stabilities and brain permeability in the future, and then we will apply other routes of administration such as i.v., i.p. or oral administration for translation to human as suggested by the reviewer.

Please add the limitations of this study. 

=> According to the reviewer’s suggestions, we added the limitations at “4. Conclusions” in the main manuscript as below.

“However, due to the low metabolic stability of the developed MFCs, there were limitations to determine the PK properties of MFCs and appropriate routes of drug administration for clinical application. Therefore, lead optimization research with the aim of improving pharmacokinetic properties should be conducted in the future based on this study, which may change the paradigm of drug development targeting multifactorial diseases.”

The model used for efficacy used was withdrawal to von Frey. Please elaborate if other tests might be required in addition to mechanical sensitivity.

=> Although mechanical sensitivity was evaluated in this study, cold allodynia evaluation such as acetone test can be considered as an additional evaluation method for temperature sensation. However, in the pain model caused by direct nerve damage applied as in this study, the mechanical allodynia evaluation method is known as the gold standard experiment.

Round 2

Reviewer 1 Report

Figure 1A and 2A still require statistical analyses by one-way ANOVA and appropriate post-hoc tests. It is not sufficient to only analyze the MPE. Please address this.

Author Response

Thank you for the comments. According to the reviewer`s suggestions, we updated the figure 1A, 2A, the figures` legends, and 3.6. Statistical analysis.